# Transient and resident pathogens: Intra-facility genetic diversity of *Listeria monocytogenes* and *Salmonella* from food production environments

**James B. Pettengill**[1]*, **Hugh Rand**[1], **Shizhen S. Wang**[1], **Donald Kautter**[2], **Arthur Pightling**[1], **Yu Wang**[1]

1 Biostatistics and Bioinformatics Staff, Office of Analytics and Outreach, Center for Food Safety and Applied Nutrition, US Food and Drug Administration, College Park, MD, United States of America, 2 Division Of Plant Products & Beverages, Office of Food Safety, Center for Food Safety and Applied Nutrition; US Food and Drug Administration, College Park, MD, United States of America

* james.pettengill@fda.hhs.gov

**Data Availability Statement:** All relevant data are within the manuscript and its Supporting Information files.

## Abstract

Food production facilities are often routinely tested over time for the presence of foodborne pathogens (e.g., *Listeria monocytogenes* or *Salmonella enterica* subsp. *enterica*). Strains detected in a single sampling event can be classified as transient; positive findings of the same strain across multiple sampling events can be classified as resident pathogens. We analyzed whole-genome sequence (WGS) data from 4,758 isolates (*L. monocytogenes* = 3,685; *Salmonella* = 1,073) from environmental samples taken by FDA from 536 U.S. facilities. Our primary objective was to determine the frequency of transient or resident pathogens within food production facilities. Strains were defined as isolates from the same facility that are less than 50 SNP (single-nucleotide polymorphisms) different from one another. Resident pathogens were defined as strains that had more than one isolate collected >59 days apart and from the same facility. We found 1,076 strains (median = 1 and maximum = 21 strains per facility); 180 were resident pathogens, 659 were transient, and 237 came from facilities that had only been sampled once. As a result, 21% of strains (180/ 839) from facilities with positive findings and that were sampled multiple times were found to be resident pathogens; nearly 1 in 4 (23%) of *L. monocytogenes* strains were found to be resident pathogens compared to 1 in 6 (16%) of *Salmonella* strains. Our results emphasize the critical importance of preventing the colonization of food production environments by foodborne pathogens, since when colonization does occur, there is an appreciable chance it will become a resident pathogen that presents an ongoing potential to contaminate product.

## Introduction

Despite the myriad guidance that exists to control the presence of foodborne pathogens (e.g., Good Manufacturing Practice (GMP), Hazard Analysis and Critical Control Point (HACCP) plans, and Food Safety Management Systems (FSMS)) [1,2] such pathogens may still be found

**Funding:** The author(s) received no specific funding for this work.

**Competing interests:** The authors have declared that no competing interests exist.

in food production environments and ultimately may result in contamination of food. The plans to ensure food safety may consist of some form of monitoring program that includes testing the environment and products for pathogens [3]. The U.S. Food and Drug Administration (FDA) performs inspections of food production facilities it regulates. Those inspections may be initiated as part of commodity-based assignments, which seek to gain insight about a particular industry; risk-based prioritization, which is the result of a focus on food-hazard pairs (e.g., frequency of outbreaks associated with a certain food type); and for-cause inspections, which are to further investigate a specific problem regarding a facility [4]. During such investigations the FDA often collects environmental samples (e.g., swabs and sponges) that are then tested in FDA laboratories to determine whether foodborne pathogens are present. The FDA may also visit the same facility over time (e.g., years) and collect samples.

There are two general patterns that can describe the presence of a pathogen within a facility's food production environment, over time. First, there is the possibility of the strain being a transient pathogen where it is not found during subsequent sampling events [1,5,6]. A sampling event here is defined as the collection of environmental samples from the facility at a specific point in time that are then tested for the presence of a foodborne pathogen (e.g., *Listeria monocytogenes* or *Salmonella enterica* subsp. *enterica*). The explanations for a negative sampling event are either the failure to detect the pathogen or the pathogen is not present in the samples. The latter may be the result of natural processes or intentional actions on behalf of the firm to eradicate pathogens.

The second general pattern involves a resident pathogen, in which the same strain is found during subsequent sampling events. This suggests that the pathogen has established itself within the environment [1,5,6]. Given the importance of controlling foodborne pathogens within food production facilities, many studies have investigated the questions of whether and how pathogens may persist in such environments. Often resident pathogens are the result of colonization of an area of the facility that is a difficult to adequately clean (e.g., cracks, junctions in the structure, drains, holes) and thus represent a niche or harborage site [7]. At such sites there is also often an accumulation of food debris or moisture that fosters the growth and persistence of pathogens such as *L. monocytogenes* and *Salmonella* [1]. Low-moisture foods and environments are known to facilitate long-term persistence of *Salmonella* [8,9]. Areas where biofilm formation is possible may also facilitate residency as biofilms increase the pathogen's tolerance to many tools used as control measures (e.g., heat, desiccation, chlorine, and antimicrobials) [10]. *L. monocytogenes* is more likely to persist in floors and drains where conditions exist that also enable biofilm formation [11]. Persistence of *L. monocytogenes* or *Salmonella* may be on the order of years [12] and has been documented in numerous environments, including those related to produce [11], cheese production environments [13], fresh-cut vegetable processing facilities [14], and crabmeat processing environments [15].

It is important to note that a potential alternative explanation to a strain being resident is reintroduction. Here, the presence of a pathogen over time within a food facility involves successful eradication from the facility, but continued contamination from external sources results in reintroduction. Although others have noted the difficulty in differentiating among the competing explanations (i.e., resident pathogen vs reintroduction) [11], the reintroduction explanation makes a number of assumptions, including elimination of the strain via intentional or natural means, persistence of the strain within the source (e.g., ingredient, material, etc.), consistency of suppliers, and subsequent successful re-colonization of the facility from which it had previously been eradicated. These numerous assumptions render the reintroduction hypothesis less parsimonious than that of residency and, while possible, we assume the results presented here of a strain being found over time within the same facility is due to residency.

In addition to the temporal requirement (e.g., present during multiple sampling events over time), differentiating between transient and resident strains requires a method for determining whether two or more isolates represent the same strain. We use the term "strain" as defined by Tenover et al. [16] to be a group of isolates from the same lower taxonomic rank (e.g., species) that can be distinguished from other isolates from the same rank based on genetic differences. Whole-genome sequencing, with its high discriminatory power and ability to differentiate at the nucleotide level, represents the best method currently available to delineate strains [17] and has been noted for its ability to characterize persistent strains [18].

Here we determine the frequencies of transient and resident pathogens across a large number of *L. monocytogenes* and *Salmonella* isolates that were collected from food production facilities within the U.S. Given there is no formal agreed upon genetic difference threshold with which to differentiate strains, we first identified such a threshold based on single nucleotide polymorphisms (SNPs) detected through comparative genome analyses. We then determined the number of different strains that were found and classified them as either transient or resident. Given the large number of facilities investigated here, our findings provide valuable insight into the magnitude of the resident pathogen problem within food production facilities.

## Materials and methods

### Data curation

We first identified all FDA *L. monocytogenes* and *Salmonella* isolates for which whole-genome sequencing had been performed and had an NCBI (National Center for Biotechnology Information) BioSample accession. Additional metadata regarding the collection of each isolate (e.g., firm ID, collection date and facility description code [e.g., fish or dairy facility]) were also extracted from an internal FDA database, as were all sample collection dates for a given facility, which was used to determine sampling events that did not return a positive result for the presence of a foodborne pathogen. Only FDA isolates from environmental samples (e.g., "environmental swab" and "environmental sponge") and from facilities located in the U.S. were included. We focused on only environmental samples because we are interested in what has directly been found in the manufacturing environment of the facility in defining transient and resident strains. We recognize this excludes other samples (e.g., product samples) that could potentially also be used to make statements about resident or transient strains. We also acknowledge that these samples are predominantly the result of inspections to assess compliance and do not represent a random sample.

This data curation resulted in a dataset of whole-genome sequence data for 4,758 isolates (*L. monocytogenes* = 3,685; *Salmonella* = 1,073) from environmental swabs taken from 536 U.S. facilities between August 2000 and March 2020 (Table 1). All sequence data analyzed here are publicly available and predominantly paired-end short-read shotgun sequence data generated using Illumina's MiSeq platform. See S1 Table for NCBI (National Center for Biotechnology Information) SRA (sequence read archive) and BioSample accessions.

**Table 1. The number of isolates (collected between August 2000 and March 2020) analyzed, and the number of different strains and resident pathogens detected.**

| Taxon | Isolates | Facilities | Strains | Facilities with Resident Pathogen | Resident Pathogens | Transient Pathogens | Strains from facilities not Revisited |
|---|---|---|---|---|---|---|---|
| *L. monocytogenes* | 3685 | 387 | 756 | 103 | 143 (23.75%) | 459 (76.25%) | 154 |
| *Salmonella* | 1073 | 151 | 320 | 23 | 37 (15.61%) | 200 (84.39%) | 83 |
| Total | 4758 | 536* | 1076 | 126 | 180 (21.2%) | 659 (78.8%) | 237 |

* Number of unique facilities.

For these 536 facilities with WGS data, we also determined the frequency of visits, time between visits, and the frequency with which a visit resulted in a positive finding (e.g., *Salmonella* or *L. monocytogenes* was detected). However, 42 of these facilities were excluded due to either insufficient information to reliably determine the complete environmental sampling history or because sampling dates fell outside the time window investigated here (i.e., only the facilities with samples collection between August 9, 2000, and March 9, 2020 were included). As a result, a total of 494 facilities were used to describe sampling (375 facilities were sampled for *L. monocytogenes*; 225 facilities were sampled for *Salmonella*); 536 facilities are included for the analysis of WGS data and identifying whether a strain was resident or not.

### Strain delineation and serotype identification

To characterize the genetic diversity among isolates within each facility, we began by generating *de novo* assemblies using the program SKESA v2.2 [19] with default settings. To estimate genetic variation among isolates from facilities with more than one isolate, we used the *de novo* assemblies and kSNP3 [20], a k-mer based variant detection program, to generate SNP matrices for isolates from a given facility. The value of k was set to 19 and the minimum fraction of isolates within which a variant had to be found to be included in the SNP matrix was 0.9 (i.e., no position in the matrix could have >10% missing data).

To determine the number of different strains within a given facility, regardless of the time over which they had persisted, we performed complete-linkage clustering using the hclust function within R [21]. We performed simulations varying the cuttree setting from 0 to 500 at increments of 5. This cuttree setting is used to delineate sub-clusters or strains in our case. Complete-linkage clustering was chosen as it produced clusters more consistent with what would be expected based on pairwise SNP distances (e.g., tree height of 50 corresponded with a 50 SNP distance threshold among samples in the sub-cluster).

It was not feasible to perform traditional serotyping on all isolates investigated in this study and, therefore, we rely on serotype prediction from the *de novo* assemblies to provide insight into patterns of resident and transient pathogens with respect to serotype. For *L. monocytogenes*, we used LisSero v0.4.1 [22], which is based on the five-locus PCR method described in Doumith [23]. That widely used five-locus serotyping schema differentiates the three major *L. monocytogenes* lineages (I, II, III) into five serogroups (serogroup I.1 includes 1/2a-3a, serogroup I.2 includes 1/2c-3c, serogroup II.1 includes 4b4d-4e, serogroup II.2 includes 1/2b-3b-7, and serogroup III includes 4a-4c). Although the method does not differentiate all serotypes within a lineage, the five serogroups are consistent with phylogenetic relationships and differentiate the most common serovars associated with foods and clinical samples [e.g., 1/2a, 1/2b, 1/2c and 4b; 23, 24]. For more detailed information on the lineages of *L. monocytogenes* we also used the program mlst [25] that uses the PubMLST [26] schema and nomenclature to provide sequence types and lineages (S2 Table). For *Salmonella*, we used SeqSero2 [27] to predict serotype based on the *de novo* assemblies. If an isolate's serogroup or serotype could not be predicted or multiple classifications were predicted for isolates from the same strain, we assigned the most abundant serogroup or serotype to all isolates from the same strain.

### Strain classification and comparison

Strains were assigned to one of three categories. 1) Resident strains were identified as those strains whose isolates were collected at least 60 days apart. 2) Transient strains were identified as those not found across multiple temporally spaced sampling events (i.e., >59 days apart) for a specific facility. 3) "strains from facilities not revisited" were identified as those from a facility that had not been sampled multiple times at least 60 days apart. Strains from a facility that was

sampled multiple times >59 days apart and were only found on the last sampling event are classified as transient.

We compared the differences between *L. monocytogenes* and *Salmonella* in how likely each was to become resident via a $x^2$-test with the *p*-value estimated from 2000 simulations. We used the paired Wilcoxon Signed-Rank test implemented in R to determine whether predicted serotypes differed in how likely strains were to be resident or transient. Non-typeable strains were not included in the test for differences among *L. monocytogenes* serotypes in how likely each was to be a resident or transient pathogen.

Fisher's exact test was used to determine whether there were differences in facility description code (e.g., a facility produces "Cheese and Cheese Products" or "Egg and Egg Products") in the relative number of transient and resident strains found. For facility description code analyses, we only included those facilities that were assigned a single description code, which avoids the issue of assigning strains to a specific type of product made at a facility with multiple description codes.

## Results and discussion

### Sampling

Sampling histories were analyzed for 494 of the 536 facilities from which we had WGS data. Of those 494 facilities, 375 were sampled for *L. monocytogenes* and 127 of those facilities were visited only once during the time window studied. The waiting time between visits for the rest of 248 facilities visited more than once ranged from 1 to 4,892 days with an average of 502 days. The number of visits to each of the 375 facilities ranged from 1 to 15, with an average of 3 per facility; the percent of visits resulting in a positive finding for *L. monocytogenes* ranged from 0% to 100%, with an average of 70.0% over all facilities. The number of environmental samples collected from each facility ranged from 1 to 15 across all visits with an average of 3 per facility (FDA samples consist of multiple subsamples); The percent of samples positive for *L. monocytogenes* ranged from 0% to 100% with an average of 68.6% over all facilities.

Of the 225 facilities included in this study where environmental sampling for *Salmonella* occurred, 82 facilities were visited only once during the time window studied. The time between visits for the rest of 143 facilities visited more than once ranged from 1 to 5,263 days with an average of 418 days. The number of visits to each of the 225 facilities ranged from 1 to 13, with an average of 3 per facility; the percent of visits yielding a detection for *Salmonella* ranged from 0% to 100% with an average of 37.5% over all facilities. The number of environmental samples collected from each facility ranged from 1 to 37 across all visits with an average of 4 per facility; the positive rate ranged from 0% to 100% with an average of 32.4% over all facilities.

### Simulations of strain definition via complete-linkage clustering

A method for assigning isolates to the same strain is important to quantifying the prevalence of resident pathogens within facilities. To accomplish this, we applied complete-linkage clustering to a genetic distance matrix among isolates from the same facility. Based on simulations, we found that the number of strains delineated was sensitive to changes in the SNP distance threshold at SNP distances less than 50 (Fig 1A). For example, a SNP distance threshold of 10 produced a mean and maximum number of strains per facility of 3.29 and 27, respectively; a SNP distance threshold of 30 produced a mean of 2.4 and maximum of 23 strains per facility. Given this behavior, we chose to delineate strains as those isolates that were within 50 SNPs of one another. Furthermore, it is important to note that the vast majority of strains have pairwise SNP distances less than that where they are actually likely to be less than 10 SNPs different (Fig 1B). This magnitude of SNP differences and threshold for strain delineation is in line with

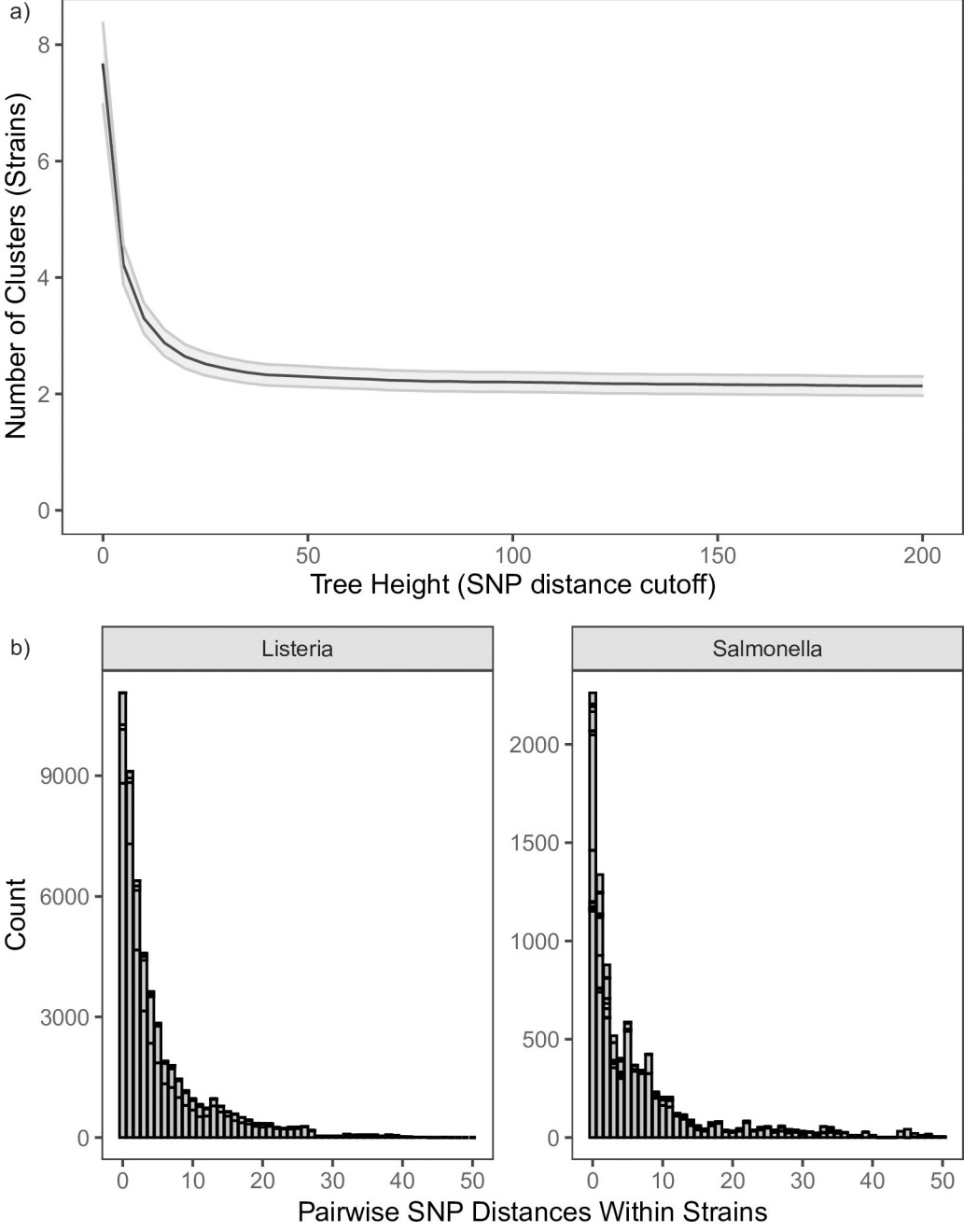

**Fig 1.** a) The number of clusters (strains) detected within each facility as a function of the tree height parameter, which corresponds to SNP distance, in the complete-linkage clustering algorithm. The mean (black line) and 95% confidence level (shaded gray region are plotted. b) Histograms of the pairwise SNP distance among isolates from the same strain at a sub-cluster height (i.e., SNP distance threshold) of 50.

what other studies have found. For example, a study of crabmeat processing environments observed that the SNP distances within isolates attributed to the same resident pathogen were on the order of a few to tens of SNP differences [15]. Additionally, the NCBI's Pathogen Detection (https://www.ncbi.nlm.nih.gov/pathogens/) platform uses a 50 SNP distance threshold for single linkage clustering.

## Resident and transient strains

We found 1,076 strains of *Salmonella* or *L. monocytogenes* across 536 facilities (median = 1 and maximum = 21 strains per facility) (Table 1). Accounting for whether a facility had been sampled multiple times and, thus, there was the possibility to detect the same strain, 237 strains came from facilities that had only been sampled once and 659 were transient. As a result, 21% (180 resident strains out of 839 [= 1,076–237]) were resident. Those 180 resident pathogens were found in 126 facilities and, thus, some facilities had multiple resident pathogens (median = 1, max = 6). As noted, an alternative explanation to residency of a pathogen is the successful eradication and subsequent reintroduction of a strain. Differentiating among those hypotheses remains an issue [e.g., 11] and additional research is necessary to gauge the frequency of reintroduction. Although reintroduction is less parsimonious, it is a possibility, and such instances would reduce the number of resident pathogens identified here.

Other studies have documented the presence of resident pathogens within food production facilities, but those have often examined a single or few facilities and did not consider data collected over a larger timescale [12,28–30]. Of exception is a study by Leong, et al. [31] that investigated 54 businesses in Ireland and found 86 different pulsotypes, of which 17 (20%) were determined to be resident, which is consistent with the observations presented here of 21% of strains being resident. The reasons for persistence within a facility are numerous, and studies have shown that the easy-to-clean food contact surfaces are less likely to have subsequent positive findings compared to harder reach areas such as cracks, drains, and internal pieces on equipment. Additionally, areas such as drains have been identified as particularly vexing to efforts to adequately control *L. monocytogenes;* prevalence in drains was nearly the same before and after control measures were implemented within smoked fish facilities [32]. Another reason is the potential route of contamination. Sauders and Wiedmann (2007) provided two reasons for why *L. monocytogenes* outbreaks may occur. One possibility is contaminated raw ingredients coming into a facility that, due to handling errors, produces a few contaminated lots but the contamination then goes away. Another possibility is facility maintenance that allows for residence to be established, and resident strains may contaminate food, potentially over a long time period. Here we found that resident strains can persist upwards of 12 years; most were found across two years (Fig 2).

## Taxon and serotype specific differences

Of the strains detected, 756 (70%) were *L. monocytogenes* and 320 (30%) were *Salmonella* (Table 1, Fig 2). We found that among resident and transient strains, *L. monocytogenes* was more likely than *Salmonella* to become a resident pathogen ($p = 0.039$, $x^2 = 4.44$).

The most frequently found *L. monocytogenes* serogroup was 1/2a, 3a with 365 strains (48%) followed by 1/2b, 3b, 7 (Table 2). Although we cannot rule out the possibility that some of those strains are 3a rather than 1/2a, the dominance of 1/2a, 3a is consistent with previous studies documenting that that 1/2a is the most abundant serogroup found in food production environments [6,24]. Although, many outbreaks and clinical infections have also historically been attributed to 4b, our results showing it to be the third most abundant serotype found supports the putative underrepresentation of that serotype within foods [33]. Interestingly, no statistical differences were detected among *L. monocytogenes* serotypes in the likelihood that strains were resident or transient (V = 10, p-value = 0.125), which may in part be explained by our low power to detect differences given there are only four different serotypes considered. However, the results here may support findings that there are not inherent genomic or phenotypic differences among the serogroups that enables a serogroup to establish and become resident within a facility compared to other serogroups [34]. Alternatively, Orsi, et al. [24] suggest

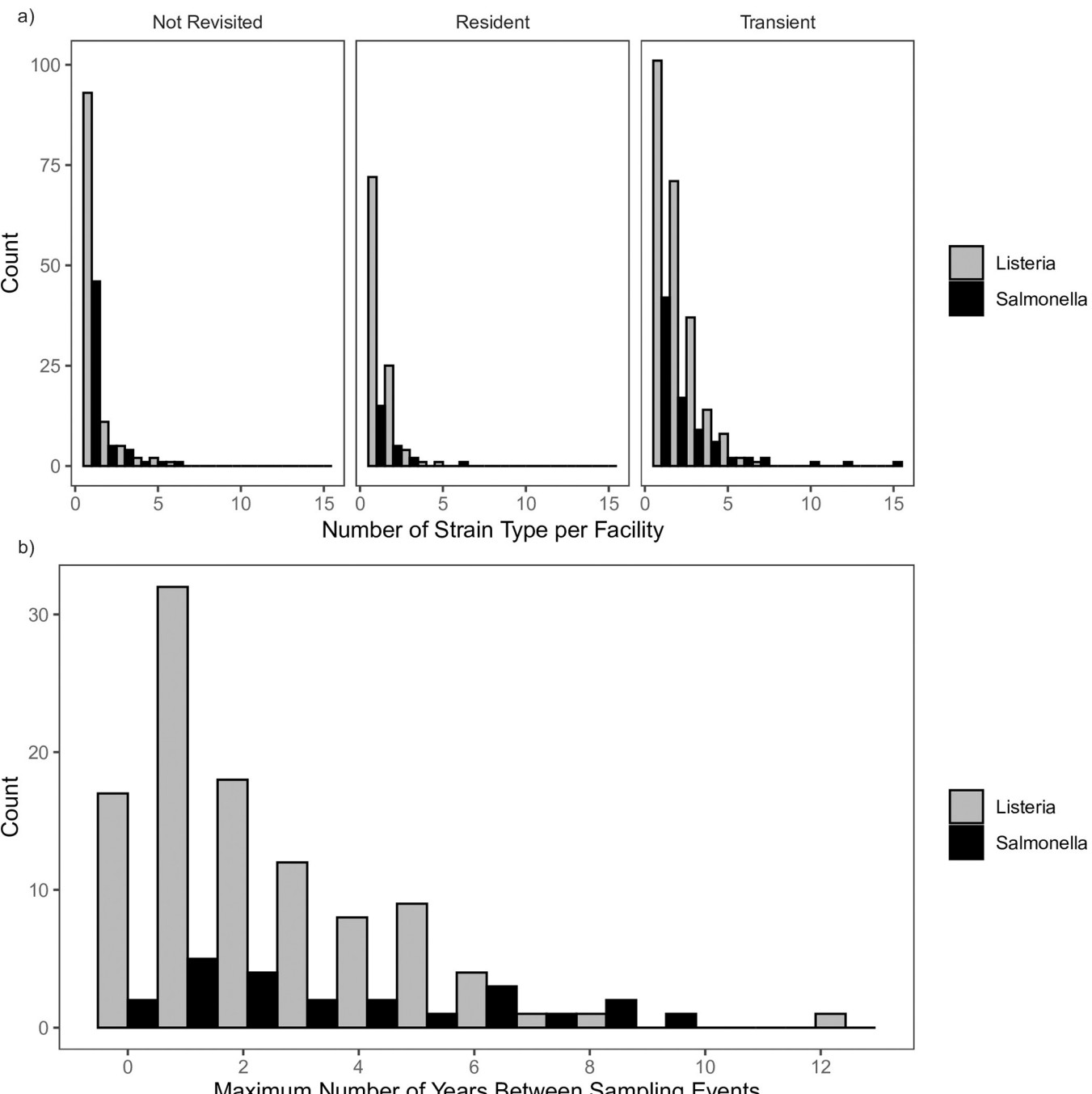

**Fig 2.** a) Histograms of the number of different strain types found per facility. b) Histogram of the maximum differences in collection dates per facility for isolates that were from resident pathogens.

that distinct features do exist among serotypes. Additionally, the dynamics of biofilm production and likelihood of a pathogen to persist are complex and vary among serotypes depending on environmental factors [35].

As for *Salmonella*, 100 serotypes were detected among the 320 strains. Serotype Enteritidis was the most frequently found, with 37 strains, followed by Senftenberg with 27 (Table 3). In

**Table 2. Distribution and classification (i.e., resident, transient, from facilities not revisited) of strains across *L. monocytogenes* serotypes.** Percentages are relative to the total number of Transient and Resident strains.

| Serotype | Transient | Resident | From Facilities Not Revisited | Total |
|---|---|---|---|---|
| 1/2a, 3a | 223 (37.04%) | 75 (12.45%) | 67 | 365 |
| 1/2b, 3b, 7 | 128 (21.26%) | 48 (7.97%) | 52 | 228 |
| 4b, 4d, 4e | 74 (12.29%) | 16 (2.65%) | 22 | 112 |
| 1/2c, 3c | 13 (2.15%) | 4 (0.66%) | 5 | 22 |
| Nontypeable | 21 (3.48%) | 0 (0%) | 8 | 29 |
| Total | 459 (76.24%) | 143 (23.75%) | 154 | 756 |

contrast to *L. monocytogenes*, statistical differences were detected among the top 10 most represented *Salmonella* serotypes in the likelihood that strains were resident or transient (V = 55, p-value = 0.002). This is likely driven by the relatively larger differences in the numbers of transient and resident strains in Enteritidis, Newport, Anatum, Mbandaka, and Muenchen. Senftenberg and Montevideo had a greater number of resident strains compared to transient strains when compared to the most abundant serotypes in terms of transient strains (e.g., Senftenberg had 14 transient strains and 7 resident strains while Enteritidis had 19 transient and 1 resident strain; Table 3).

## Resident strains and facility description codes

There were differences in the number of strain types associated with the different facility description codes (e.g., dairy facility or seafood facility) for *L. monocytogenes* (p-value = 0.042) (Table 4). For example, "Fishery/Seafood Products" and "Ice Cream and Related Products" tended to harbor larger fractions of resident strains than "Cheese and Cheese Products" or "Vegetables and Vegetable Products". The differences in the number of strain types associated with facility type for *Salmonella* was not statistically supported (p-value = 0.328), but the distribution of strain types across facilities with positive *Salmonella* findings illustrates the small fraction of *Salmonella* resident strains compared to what is seen for *L. monocytogenes* (Table 1).

We also observed differences in the numbers of strains of *L. monocytogenes* and *Salmonella* among different facility description codes (p-value = 0.001) which are consistent with the published literature. For example, from a comparison of expert elicitations and risk-based studies,

**Table 3. Distribution and classification (i.e., resident, transient, or not revisited) of strains from the 10 most abundant *Salmonella* serotype detected.** Percentages are relative to the total number of Transient and Resident strains.

| Serotype | Transient | Resident | From Facilities Not Revisited | Total |
|---|---|---|---|---|
| Enteritidis | 19 (16.1%) | 1 (0.85%) | 19 | 37 |
| Senftenberg | 14 (11.86%) | 7 (5.93%) | 14 | 27 |
| Newport | 12 (10.17%) | 0 (0%) | 12 | 20 |
| Anatum | 10 (8.47%) | 2 (1.69%) | 10 | 13 |
| Mbandaka | 10 (8.47%) | 1 (0.85%) | 10 | 13 |
| Muenchen | 10 (8.47%) | 0 (0%) | 10 | 11 |
| Cubana | 6 (5.08%) | 3 (2.54%) | 6 | 10 |
| Infantis | 7 (5.93%) | 1 (0.85%) | 7 | 10 |
| Montevideo | 5 (4.24%) | 3 (2.54%) | 5 | 10 |
| Heidelberg | 6 (5.08%) | 1 (0.85%) | 6 | 9 |
| Total | 99 (83.9%) | 19 (16.1%) | 99 | 160 |

**Table 4. The number of strain types among the top 10 most abundant facility description code types.** Facilities assigned multiple description code types were excluded.

| Facility Description Code | Listeria | | | Salmonella | | | Total |
|---|---|---|---|---|---|---|---|
| | From Facilities Not Revisited | Transient | Resident | From Facilities Not Revisited | Transient | Resident | |
| Fishery/Seafood Products | 22 | 80 | 28 | 0 | 1 | 0 | 131 |
| Cheese and Cheese Products | 13 | 42 | 7 | 1 | 0 | 0 | 63 |
| Egg and Egg Products | 0 | 0 | 1 | 15 | 32 | 2 | 50 |
| Ice Cream and Related Products | 8 | 17 | 12 | 1 | 0 | 0 | 38 |
| Nuts and Edible Seeds | 0 | 0 | 0 | 10 | 24 | 2 | 36 |
| Vegetables and Vegetable Products | 2 | 19 | 3 | 9 | 2 | 0 | 35 |
| Fruit and Fruit Products | 7 | 5 | 0 | 6 | 9 | 0 | 27 |
| Milk, Butter, and Dried Milk Products | 0 | 0 | 0 | 2 | 12 | 4 | 18 |
| Bakery Products, Doughs, Bakery Mixes, and Icings | 6 | 5 | 3 | 0 | 0 | 0 | 14 |
| Multiple Food Dinners, Gravies, Sauces, and Specialties (Total Diet) | 5 | 3 | 1 | 0 | 0 | 0 | 9 |

the most hazardous foods regarding Listeriosis were deli meats, dairy products and seafood, but of note is the elicitation also identified produce as being important [36]. Here, "Fishery/ Seafood Products", "Cheese and Cheese Products", and produce ("Vegetables and Vegetable Products" and "Fruit and Fruit Products) facilities had a high number of different strains, which is congruent with the discussion in Todd and Notermans [36]; however, this may be a sampling artifact, since those facilities were also the most frequently sampled. Of note is the lack of *L. monocytogenes* strains found in facilities with description code "Milk, Butter, and Dried Milk Products", which is typically classified as dairy. This may suggest that *L. monocytogenes* is more specific to cheese products than the milk aspect of the general dairy classification, but this result is also likely due in part to differences in sampling priorities (i.e., *Salmonella* environmental monitoring is done for dried dairy products while *L. monocytogenes* is not). *Salmonella* strains were most frequently found in "Egg and Egg Products" facilities, followed by "Nuts and Edible Seeds" facilities (Table 4). Although both commodity types are known to be associated with *Salmonella* contamination [37,38], among facilities assigned a single description code "Egg and Egg Products" facilities were sampled less than "Nuts and Edible Seeds," yet there are more strains found in the former suggesting *Salmonella* is more likely to be found in "Egg and Egg Products" facilities. Within "Egg and Egg Products" facilities, Enteritidis was the dominant serotype found, followed by Heidelberg, which is to be expected based on previous research [e.g., 39]. For "Nuts and Edible Seeds" facilities, serotypes Montevideo, Newport, Senftenberg, and Tymphimurium each had three strains found (not shown). *Salmonella* strains were also prominent in fruit and vegetable facilities (Table 4).

## Conclusions

There are numerous steps along the food production chain where products could become contaminated with a foodborne pathogen. Here, we have investigated one potential source of contamination–the facility in which ingredients are processed and products are manufactured. We found 1,076 strains across 536 facilities and, most importantly, that 21% of strains within facilities that were sampled over time were found to be resident pathogens. This pattern across such a large number of facilities, isolates, and strains highlights the importance of inhibiting introduction into or ensuring efficient eradication of pathogens in the production environment, since not doing so can lead to a potentially long-term problem associated with a resident

pathogen and managing the constant risk a resident strain represents to producing food free of environmental pathogens such as *L. monocytogenes* and *Salmonella*.

## Supporting information

**S1 Table. NCBI (National Center for Biotechnology Information) SRA (sequence read archive) and BioSample accessions for sequence data analyzed.**
(XLSX)

**S2 Table. Lineages and sequence types of *L. monocytogenes* as predicted with the program mlst (25) that uses the PubMLST (26) schema.**
(XLSX)

## Acknowledgments

We appreciate insightful feedback from A. Dwarka and J. Scott on the manuscript. We acknowledge those responsible for collecting and sequencing the samples analyzed here, which was done by several groups, including the US FDA Office of Regulatory Affairs (ORA) and US FDA Center for Food Safety and Applied Nutrition's Office of Regulatory Science (ORS) and Office of Compliance (OC).

## Author Contributions

**Conceptualization:** James B. Pettengill, Donald Kautter, Arthur Pightling, Yu Wang.

**Data curation:** Shizhen S. Wang, Yu Wang.

**Formal analysis:** James B. Pettengill, Shizhen S. Wang.

**Methodology:** Hugh Rand.

**Visualization:** James B. Pettengill.

**Writing – original draft:** James B. Pettengill, Arthur Pightling.

**Writing – review & editing:** Hugh Rand, Shizhen S. Wang, Donald Kautter, Arthur Pightling, Yu Wang.

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
