## [Decision Letter · Decision Letter 0]

5 Jan 2022

PONE-D-21-35348Transient and resident pathogens: intra-facility genetic diversity of Listeria monocytogenes and Salmonella from food production environmentsPLOS ONE

Dear Dr. Pettengill,

Thank you for submitting your manuscript to PLOS ONE. After careful consideration, we feel that it has merit but does not fully meet PLOS ONE’s publication criteria as it currently stands. Therefore, we invite you to submit a revised version of the manuscript that addresses the points raised during the review process.

We look forward to receiving your revised manuscript.

Kind regards,

Carlo Spanu, Ph.D.

Academic Editor

PLOS ONE

3. PLOS requires an ORCID iD for the corresponding author in Editorial Manager on papers submitted after December 6th, 2016. Please ensure that you have an ORCID iD and that it is validated in Editorial Manager. To do this, go to ‘Update my Information’ (in the upper left-hand corner of the main menu), and click on the Fetch/Validate link next to the ORCID field. This will take you to the ORCID site and allow you to create a new iD or authenticate a pre-existing iD in Editorial Manager. Please see the following video for instructions on linking an ORCID iD to your Editorial Manager account: https://www.youtube.com/watch?v=_xcclfuvtxQ.

Reviewers' comments:

Reviewer's Responses to Questions

**Comments to the Author**

1. Is the manuscript technically sound, and do the data support the conclusions?

Reviewer #1: Partly

Reviewer #2: Yes

2. Has the statistical analysis been performed appropriately and rigorously? 

Reviewer #1: I Don't Know

Reviewer #2: Yes

3. Have the authors made all data underlying the findings in their manuscript fully available?

Reviewer #1: No

Reviewer #2: Yes

4. Is the manuscript presented in an intelligible fashion and written in standard English?

Reviewer #1: Yes

Reviewer #2: Yes

5. Review Comments to the Author

Reviewer #1: The study describes a large-scale analysis on Listeria monocytogenes and Salmonella isolates collected from food industry by the FDA, where the authors perform analyses on whether the strains are resident or transient. I welcome such a large-scale study, in comparison with many small and often single-plant studies, as this may allow uncovering larger patterns. However, I think that the total lack of information about the sampling (no. of samples, no. of samplings, interval between samples etc.) makes it difficult for the reader (and for me as a referee) to evaluate the results and the justification of grouping strains as resident/transient. Also, the author’s fail to discuss how variation in these (sampling) parameters may influence their analysis, which they in my opinion may do. The grouping of strains according to serotype and not CC/ST makes the study less relevant for readers.

Introduction: The introduction is lacking background on studies of persistence of Salmonella in the food industry. Please include references/examples from other studies to justify the relevance of your hypothesis that Salmonella may persist in the food industry.

Line 67-73. While I realize that you need to exclude Reintroduction to perform your Resident/Transient analyses, your arguments about why it is not relevant are rather weak. Raw materials are a relevant source of introduction of pathogens and if raw materials contain a “resident” strain this may lead to overestimation of resident strains colonizing plants. Of course, food safety regulations should limit such reintroduction to hygienic zones in the factories, but results from FDA inspections reveal that it is not uncommon with sanctions for improper hygiene, lack of zoning etc. I think you should include something about how Reintroduction may have influenced your analysis in the discussion.

Introduction, Methods, Results: Serotyping? In a paper based on WGS, why do you not refer to CC or ST? At least for Listeria, almost all papers based on typing by sequencing use this type of grouping, and not serotyping. Other papers suggest that some CC/ST (e.g. 121, 9) to be dominating as resident strains, and I think it would have been interesting for readers with an analysis of which CC/ST that dominated among resident strains in your analysis, and a comparison with other studies. As you have the sequences this should be straightforward to do. This can be in addition, and not instead of the serotyping part.

Materials and Methods:

I think the authors should include information about the variation among plants/facilities in number of visits, number of samples taken, and interval between samplings. The lack of such information makes it difficult for the reader to evaluate the results. I acknowledge that there may be huge variations in these parameters but information about the distribution of this variance should be presented. Based on the information presented in the paper it does not exclude the possibility that e.g. 50% of the plants was visited twice and one sample taken at each sampling. Obviously this is a stupid assumption and even this stupid referee realizes this is not the case, but as you provide absolutely no information about the sampling, the results are very difficult to evaluate for the reader, and it is obvious that number of samplings and samples as well as the time interval between samplings can influence the analysis leading to conclusions on whether a strain is resident or transient.

Results and discussion:

As commented above; I think the authors should include information about CC/ST of the strains (at least for Listeria) and compare their results on CC/ST with other studies.

As commented above there is a need to provide more information about the variations in the samplings, and to discuss how this may have influenced your results.

I suggest to include more information about and compare with other studies on persistence/resident/transient strains. How common is persistence/resident strains. Are your results in line with other results. Such an evaluation/comparison is especially lacking for Salmonella.

Conclusions: References are usually not used in conclusions I suggest to exclude the references, and shorten the conclusion to focus on your own findings.

Reviewer #2: Summary

The authors present a SNP-based analysis of Listeria and Salmonella collected by government sampling of US food processing facilities. They use this analysis to differentiate persistent from sporadic strains by a SNP threshold, and then describe how those frequencies change given various important categories.

General Comments

Overall, this is a useful paper because only the US government would have access to the metadata for Genomtrakr isolates to do this type of analysis. From that unique position, the analysis and results seem appropriate. There is one major comment, and request for including more data:

L97: This ‘denominator’ data of all sampling events that did not include a positive sample is very useful data that could only be analyzed by the FDA, academics cannot see such data. Therefore, the reviewer would really appreciate seeing this data summarized in Table 1. Though I understand there may be legal reasons it could not be included. This presentation of negative sampling events would help support the discussion around Table 4 – facility types. It might even allow statistical control for sampling frequency.

A few additional comments follow that would improve the analysis or presentation.

Line Item Comments

L27. Clarify if 50 SNPs also applies to transient strains. Maybe: strained defined as …, and strains called resident if …

Fig 1a. Useful primary data, but I would suggest plotting the median line and a quantile range. This is more consistent with the summary statistics presented in the abstract, and more informative when there is heavy skew in the data.

Fig 2a. I would like to see statistics if the distribution of counts are different between the 3 categories. Transient does appear to have a longer tail, but that could simply be because of the large number of observations.

222. No difference in serotypes likelihood of resident or transient. OK. One additional alternative explanation could be that there is already significant selection pressure for a strain to even be transient. And therefore there is not additional genetic selection around persistence (and it might be more about location and external factors like cleaning).

6. PLOS authors have the option to publish the peer review history of their article (what does this mean?). If published, this will include your full peer review and any attached files.

Reviewer #1: No

Reviewer #2: **Yes: **Matthew J Stasiewicz

---

## [Editor Report · Decision Letter 1]

1 May 2022

Transient and resident pathogens: intra-facility genetic diversity of Listeria monocytogenes and Salmonella from food production environments

PONE-D-21-35348R1

Dear Dr. Pettengill,

We’re pleased to inform you that your manuscript has been judged scientifically suitable for publication and will be formally accepted for publication once it meets all outstanding technical requirements.

Kind regards,

Carlo Spanu, Ph.D.

Academic Editor

PLOS ONE
---

## [Editor Report · Acceptance letter]

24 Aug 2022

PONE-D-21-35348R1 

Transient and resident pathogens: intra-facility genetic diversity of *Listeria monocytogenes* and *Salmonella* from food production environments  

Dear Dr. Pettengill:

I'm pleased to inform you that your manuscript has been deemed suitable for publication in PLOS ONE. Congratulations! Your manuscript is now with our production department. 

Kind regards, 

on behalf of

Dr. Carlo Spanu 

Academic Editor

PLOS ONE